# Does a rise in BMI cause an increased risk of diabetes?: Evidence from India

**Shivani Gupta**[ID]ᵒ, **Sangeeta Bansal**＊ᵒ

Centre for International Trade and Development, School of International Studies, Jawaharlal Nehru University, New Delhi, India

ᵒ These authors contributed equally to this work.
＊ sangeeta.bansal7@gmail.com

**Data Availability Statement:** Data underlying the study was collected from: Demographic and Health Surveys (2018), Data Collection, Survey for India, NFHS. URL: https://dhsprogram.com/data/. The authors confirm they did not have any special

## Abstract

### Background

Overnutrition increases the risk of diabetes. Evidence on the causal impact of overnutrition on diabetes is scarce for India. Considering a representative sample from India, this study examines the causal effect of a rise in the Body Mass Index (BMI) of an individual on the likelihood of being diabetic while addressing the issue of unobserved endogeneity between overnutrition and diabetes.

### Methods

The study considers individual level data from Demographic and Health Surveys (DHS) of India, namely, National Family Health Survey (NFHS) for the year 2015–16. The NFHS is a large-scale, multi-round survey conducted in a representative sample of households throughout India. The survey covers females having age 15–49 years and males having age 15–54 years. The instrument variable approach is used to address the potential endogeneity in the relationship between BMI and diabetes. We instrument BMI of an individual by BMI of a non-biologically related household member. Ordered Probit, Probit and IV-Probit models are estimated using two alternative definitions for measuring diabetes–self-reported diabetes status and blood glucose levels (ordinal measure).

### Results

The coefficients obtained from the Ordered Probit and Probit models are much smaller than those estimated by an IV-Probit model. The latter estimates the causal impact of a rise in BMI on diabetes by taking into account the effect of the unobserved genetic and other related factors. The likelihood of being diabetic is twice or more among the overweight and obese individuals as compared to non-overweight individuals in all the specifications. With a unit increase in BMI the probability of being diabetic increases by about 1.5% among overweight and obese individuals and by 0.5% among the non-overweight individuals in the IV-Probit model. Similar results from the Ordered Probit model show that on average, the overweight and obese individuals experience about 0.2% increase in the probability of being diabetic and about 0.4% increase in the probability of being prediabetic.

access to this data that other researchers would not have.

**Funding:** The author(s) received no specific funding for this work.

**Competing interests:** The authors have declared that no competing interests exist.

## Conclusion

Our study demonstrates that the likelihood of being both prediabetic and diabetic is higher among the overweight and obese individuals as compared to the non-overweight individuals. We also find that the level of risk of being prediabetic or diabetic differs across gender, wealth quintiles and regions and the effects are more severe among population in the urban areas, belonging to the richest wealth quintile and men. Our findings have significant implications for the policy formulation as diabetes has a substantial health and economic burden associated with it. Future studies may investigate the effect of abdominal obesity on diabetes.

## Introduction

The rise in diabetes prevalence during the past decade has begun to pose a new challenge to the health policy makers in India. In 2017, about 72 million people (8.8% of the total population having age 18 years or above) and 20% of the urban population was diabetic in India (International Diabetes Federation (IDF)) [1]. According to Diabetes Foundation of India, people suffering from diabetes are likely to go up to 80 million by 2025, making India the 'Diabetes Capital' of the world [2]. Analysing National Family Health Survey (NFHS) [3] data for the increase in the diabetes prevalence in India over a ten-year period (2005 to 2015), we find that diabetes prevalence has doubled in both rural as well as urban areas and there has been a considerable increase in almost every state.

Overnutrition has been found to be a major risk factor for a number of diseases such as diabetes, hypertension, heart diseases, certain type of cancers, etc. [4,5,6]. Overnutrition is one of the potential factors that may generate insulin resistance, which in turn may increase the sugar or glucose content in the blood leading to diabetes [7]. Using a prospective cohort study on women in the United States, Colditz et al. [5] find that the risk of diabetes is increasing in Body Mass Index (BMI). The study by Huffman et al. [4] finds similar results among married women in Delhi, India. Other factors that may lead to diabetes include smoking, alcohol consumption, high sugar intake, genetic predisposition, etc. [8,9,10].

India is going through a nutritional transition brought about by a rapid emergence of overnutrition. The rising overnutrition may have a relationship with the growing diabetes problem in India. Overnutrition is associated with the increased risk of mortality and co-morbidities [11,12]. Asian population faces this risk even at lower BMI values. The risk of chronic conditions is higher among Asian population due to increased susceptibility towards non-communicable diseases (NCDs) even at lower BMI levels as compared to the population in the European countries and the United States [13,14,15].

Many studies including Gray et al. [13] and Sepp et al. [16] have estimated the relationship between overnutrition and NCDs for European countries. These studies are based on small sample sizes and the results may not be representative for the entire population. Further, much of the evidence on the link between overnutrition and NCDs comes from the high-income countries [17,18,19]. The findings from these studies cannot be extrapolated for the Indian population due to regional differences in the body types and distribution of body fat. South Asian population is found to have a higher abdominal obesity as compared to the population in the European regions, therefore, the susceptibility towards certain types of diseases, such as diabetes, may vary across these regions even if the BMI values are comparable) [20,15].

The evidence on the effect of BMI on diabetes for Indian population is limited. Huffman et al. [4] consider a cohort sample of 1100 women in South Delhi and show that an increase in

BMI has a statistically significant impact on diabetes among married women in Delhi, India. The study by Ramachandran et al. [21] finds a positive association between diabetes and BMI for the urban population across six cities in India using the National Urban Diabetes Survey. None of the studies, however, has considered WHO Asian BMI classification, which defines an individual having BMI $\geq 23$ kg/m$^2$ as overweight or obese, to examine the effect of overnutrition on diabetes in India.

The objective of this study is to examine the effect of overnutrition on diabetes in India. We investigate the causal effect of an increase in BMI on the likelihood of suffering from diabetes using an individual level nationally representative data set for India. The novel contribution of the study is that it addresses potential endogeneity arising from the unobserved genetic and other related factors while estimating the effect of BMI on diabetes. BMI of an individual is likely to be correlated with the omitted determinants of his/her diabetes status. These omitted variables could be related to the individual's genetic and non-genetic predisposition towards overweight and obesity as well as diabetes. We address this issue by using an instrumental variable approach and instrument BMI of the individual by BMI of a non-biologically related household member. The BMI of a non-biologically related household member is correlated with the common household environment but there is no reason to believe that it will systematically affect the individual's predisposition towards diabetes. We also control for several covariates on individual characteristics, household characteristics and behavioural risk factors such as tobacco and alcohol consumption, eating habits, etc. We extract individual level data from the fourth round of NFHS for the year 2015–16. NFHS is a nationally representative data set.

We are interested in estimating the change in the probability of being diabetic with a unit gain in BMI, and comparing this effect across non-overweight and overweight or obese population. For this comparison, we apply both WHO International BMI classification, which defines an individual having BMI $\geq 25$ kg/m$^2$ as overweight or obese, and WHO Asian BMI classification, which defines an individual having BMI $\geq 23$ kg/m$^2$ as overweight or obese. One may expect the urban population and the population belonging to the higher wealth quintiles to face a higher risk of diabetes due to the lifestyle related factors and increased access to calorie dense foods across different subpopulations. Reviewing the available literature, Olinto et al. [22] find that socioeconomic status in terms of higher income and wealth are associated with higher obesity among men. The socioeconomic status and urban lifestyle factors may affect diabetes status through higher BMI levels, therefore, we also examine the heterogeneity in the effect of BMI on diabetes across different subgroups of the population based on gender (male and female), regions (rural and urban) and wealth quintiles (poorest and richest). Based on the discussion above, we test the following hypotheses:

*Hypothesis 1*: An increase in BMI increases the risk of diabetes and prediabetes.

*Hypothesis 2*: This risk is higher for overweight and obese population, i.e., with an increase in BMI, the likelihood of being diabetic and prediabetic increases more for an overweight or an obese individual as compared to a non-overweight individual.

*Hypothesis 3*: With an increase in BMI, population belonging to the higher wealth quintiles is more likely to be prediabetic and diabetic as compared to the population among lower wealth quintiles.

*Hypothesis 4*: With an increase in BMI, population living in the urban areas is more likely to be prediabetic and diabetic as compared to the population living in the rural areas.

## Methods

IDF has identified physical inactivity, consumption of unhealthy foods and sedentary lifestyle as factors that influence diabetes. A rise in BMI of an individual caused by changes in any of these

factors is likely to increase his/her susceptibility towards higher blood glucose levels) [23,16]. It is possible that an increase in BMI raises an individual's blood glucose to the levels which are not high enough to be characterised as diabetic, but can be characterised as prediabetic (defined below). If adequate measures are not taken to control the rising blood glucose levels, the individual may subsequently become diabetic with a further rise in the blood glucose levels.

We use two alternative measures for our outcome variable, diabetes status of individuals,–self-reported diabetes status and blood glucose levels. Self-reported diabetes status takes value 1 if an individual is diabetic and 0 otherwise. For the second measure, we assign ordinal values to the blood glucose levels by dividing them into three mutually exclusive categories. The blood glucose level measures the amount or concentration of the glucose in a blood sample as milligrams per decilitre (mg/dl). Following the random glucose/sugar test, we have the following three categories for the blood glucose levels:

i. Less than or equal to140 mg/dl corresponds to low or moderate blood glucose–*Normal Blood Glucose Levels*

ii. Between 141 and 200 mg/dl corresponds to high blood glucose–*Prediabetes*

iii. Greater than 200 mg/dl corresponds to very high blood glucose–*Diabetes*

In the ordinally defined blood glucose levels, we assign value 0 to normal blood glucose levels, 1 to prediabetes and 2 to diabetes. Our second measure enables us to quantify the effect of a rise in BMI on both diabetes as well as prediabetes.

While the ordinal measure tests all the hypotheses stated in the introductory section for both prediabetes and diabetes, the self-reported diabetes status measure tests these hypotheses for diabetes only. We test the third and fourth hypotheses for the full data sample as well as for a sub-sample comprising of overweight or obese population. These hypotheses are tested using both WHO International and WHO Asian classification of BMI to identify an individual as overweight or obese.

Our main explanatory variable of interest is the BMI of an individual. We control for a rich set of covariates both at the individual level as well as at household level that are likely to affect the risk of diabetes. In addition, we control for the state fixed effects. Individual characteristics include age, gender, educational attainment, behavioural risk factors and eating habits. Behavioural risk factors controlled for in our regressions include a comprehensive set of variables that measure tobacco consumption of an individual such as–smoking cigarette, smoking pipe, chewing tobacco, snuffing, smoking cigar, chewing paan, gutkha, paan with tobacco, etc., and alcohol consumption. These risks factors are likely to affect blood glucose levels and diabetes status of individuals. Available literature suggests that smoking elevates the risk of diabetes. Smoking generates insulin resistance leading to the increased risk of diabetes [24,8]. Moderate consumption of alcohol may reduce the risk of diabetes [10,9] while binge drinking may increase this risk [9,25].

Eating habits are also an important factor affecting susceptibility to diabetes. We consider daily or weekly consumption of fried foods and aerated drinks to capture eating habits. These variables also indicate consumption preferences. Food habits such as consumption of aerated drinks, fast-foods, fried foods, etc., increases the risk of obesity and insulin resistance [26,27,28]. Gulati and Misra [29] find that an increase in per capita sugar consumption leads to development of insulin resistance, abdominal adiposity and risk of diabetes.

In addition to the above, we also control for household characteristics such as wealth quintile, family structure (nuclear or joint), region (rural or urban), religion, caste, availability of health insurance, whether the household belongs to below poverty line and other covariates (A complete list of variables is provided in the data section).

Although we control for a large number of covariates, we still expect unobserved genetic and other related factors to affect relationship between BMI and diabetes. Genetic factors may influence both BMI and diabetes status of an individual. An individual with a family history of diabetes is more likely to develop diabetes even without being overweight or obese [15,30]. To address potential endogeneity due to unobserved genetic and other related factors in the form of Omitted Variable Bias (OVB), we resort to an Instrument Variable Approach. Endogeneity issue is elaborated later in this section.

## Body Mass Index and self-reported diabetes status: Probit and IV-Probit model

We estimate a Probit model having self-reported diabetes status as the binary outcome variable. The following model, having $D_i^*$ as the dependent variable, is estimated:

$$D_i^* = \beta' X_i + v_i \quad (1)$$

where,

$$D_i = \begin{cases} 0 \ \text{if individual is non} - \text{diabetic}, \\ 1 \ \text{if individual is diabetic}. \end{cases} \quad (2)$$

$i = 1, 2, \ldots, n$, represents $i^{th}$ individual;

$D_i^*$ represents latent selection variable for self-reported diabetes status of $i^{th}$ individual and is unobserved; $X_i$ represents vector of controls including BMI for $i^{th}$ individual; $v_i$ represents error term and is assumed to be independent of $X_i$ and has a standard normal distribution.

We estimate a binary response model, in which a non-linear function, $\Phi(.)$, which is a standard normal cumulative distribution function in case of Probit model, is applied to the response function. For estimating binary or ordinal response models, Maximum Likelihood Estimation (MLE) is used. We first estimate the Probit model assuming that there are no unobserved factors that affect both BMI and self-reported diabetes status of an individual, that is, $Cov(X_i, v_i) = 0$. We estimate the average marginal effects of BMI on self-reported diabetes status. We further examine if the change (or increase) in probability of being diabetic with a unit increase in BMI is higher among the overweight or obese individuals as compared to the non-overweight individuals, i.e.,

$$\left[ \frac{\partial P(D=1|X)}{\partial BMI}; \ if \ BMI \geq 25 \ kg/m^2 \right] > \left[ \frac{\partial P(D=1|X)}{\partial BMI}; \ if \ BMI < 25 \ kg/m^2 \right] > 0 \quad (3)$$

Unobserved genetic and other related factors influence both diabetes as well as overweight or obesity status of an individual, thereby, causing endogeneity resulting from OVB. This could result into a biased estimate of $\beta$. Our sample data provides self-reported values for the diabetes status of individuals, introducing another source of endogeneity in the form of measurement error. Although in case of a large dataset, the measurement error in the dependent variable does not bias the estimates [31]. We resort to an instrumental variable estimation which addresses the endogeneity caused by both OVB and measurement error.

We instrument BMI of an individual using BMI of a non-biologically related household member. Specifically, we use BMI of his/her spouse, $BMI^S$, as an instrument. The instrument must fulfil the following two requirements [32]:

i.  BMI of a non-biologically related household member, BMI of individual's spouse, must be uncorrelated with the unobserved factors that explain variations in the diabetes status of an

individual, i.e., the instrument must be uncorrelated with the error term in Eq (1):

$$Cov(BMI^S, v) = 0 \qquad (4)$$

ii. Instrument must be correlated with the BMI of individual, in other words, instrument must be powerful:

$$Cov(BMI^S, BMI) \neq 0 \qquad (5)$$

Common household factors may affect BMI of all residing individuals in a similar way due to shared family or household environment [33,34]. Studies have also documented the similarities in BMI movements among married couples [35,36,37]. Therefore, we expect the BMI of an individual and BMI of his/her spouse to be correlated.

For BMI of the spouse to be a valid instrument it should not have an independent effect on the diabetes status of the individual. BMI of an individual's spouse is likely to be uncorrelated with the unobserved genetic factors that affect the diabetes status of the individual. However, it is possible that the common household factors which affect BMI of the individual may also affect his diabetes status. Therefore, we control for several variables on the household characteristics in our model.

We estimate an IV-Probit model with the first stage equation as:

$$BMI_i = \delta_0 + \delta_1 BMI_i^s + \delta_2 x_i + \eta_i \qquad (6)$$

where, $BMI_i$ represents BMI of the $i^{th}$ individual; $BMI_i^s$ represents BMI of $i^{th}$ individual's spouse, $x_i$ represents vector of controls that include all exogenous variables of the second stage regression, and $\eta_i$ is the error term that has a standard normal distribution, $N(0,1)$.

The second stage regression includes predicted values of BMI obtained from estimation of Eq (6) in the place of the actual values of BMI as explanatory variable in Eq (1).

## Body Mass Index and blood glucose levels: Ordered Probit model

Given that the second indicator of diabetes status is a categorical variable and has more than two ordered categories, we estimate an Ordered Probit Model [38,39,40]. Following the methodology described above with the dependent variable now being, $BG_i^*$:

$$BG_i^* = \alpha' X_i + \varepsilon_i \qquad (7)$$

Blood glucose levels (dependent variable) are sorted into 3 categories:

$$BG_i = \begin{cases} 0 \ if \ BG_i^* \leq \mu_0 \\ 1 \ if \ \mu_0 < BG_i^* \leq \mu_1 \\ 2 \ if \ \mu_1 < BG_i^* \end{cases} \qquad (8)$$

where $BG_i$ represents the observed blood glucose levels for $i^{th}$ individual. The $\mu_j$'s are threshold coefficients or cut-off points. Here, $BG_i = j \Leftrightarrow \mu_{j-1} < BG_i^* \leq \mu_j$; $j = 0, 1, 2$ and, $\mu_{-1} = -\infty$ and $\mu_2 = +\infty$. We estimate the probability for an individual belonging to one of the $j$ categories:

$$P(\mu_{j-1} < BG_i^* \leq \mu_j) = \Phi(\mu_j - \alpha' X_i) - \Phi(\mu_{j-1} - \alpha' X_i) \qquad (9)$$

We estimate the above defined model using MLE. As per hypothesis 2, we expect that with a rise in BMI an overweight or obese individual is more likely to be diabetic and prediabetic. In

other words, the increase in the probability of being diabetic and prediabetic with a unit increase in BMI is higher among the overweight or obese individuals as compared to the non-overweight individuals:

$$\left[\frac{\partial P(BG = j|X)}{\partial BMI}; \ if \ BMI \geq 25 \ kg/m^2\right] > \left[\frac{\partial P(BG = j|X)}{\partial BMI}; \ if \ BMI < 25 \ kg/m^2\right] > 0; j$$
$$= 1, 2 (10)$$

## Data

The study extracts individual level data from the fourth round of NFHS for the year 2015–16 provided by Demographic and Health Surveys (DHS). The NFHS is a large-scale, multi-round survey conducted in a representative sample of households throughout India. This survey has rich information on household characteristics, individual characteristics–age, education, anthropometry, diseases and related sufferings, etc. While the survey reports the measured levels of blood glucose (our health outcome variable), the diabetes status is self-reported. The survey covers females in the age group 15–49 years and males in the age group15-54 years. We extract individual level data from three different Stata format data files published by DHS, namely, Household Member Recode, Individual Recode (Women's Recode) and Men's Recode and merged these files into one. Our analysis considers all 36 states and union territories of India. The list of variables included in the study along with their definitions is provided in S1 Table. In our sample, we include all the observations that report BMI, and either self-reported diabetes status or blood glucose levels. This gives us a total sample size of about 0.8 million observations.

In the IV-Probit model, we limit our sample to the individuals who are married and are currently living in the same household. Since NFHS provides the relationship data for each individual in terms of their relationship to the head of the household, our sample further gets restricted to married couples living together in the same household one of whom is the head of the household.

## Results

### Descriptive statistics

S2 Table presents the descriptive statistics for the entire sample. The mean blood glucose level for the total sample is 104.7 mg/dl. About 1.5% of individuals in our sample are diabetic based on the self-reported diabetes status. This is lower than the estimates given by IDF (8.8% for year 2017). This could be due to various factors such as individuals not been aware of their diabetes status, differences in age groups considered for measuring the diabetes prevalence, and also the year of sample data. Our estimate is based on age group 15–49 years for females and 15–54 years for males while IDF estimate for diabetes is for 20–79 years age group. Diabetes prevalence is expected to increase with age. Based on blood glucose levels, 94% individuals have normal blood glucose, about 5% are prediabetic and about 1% are diabetic. Blood glucose levels of some diabetic individuals could be regulated via use of medicines. The mean BMI is 21.71 kg/m$^2$ indicating that on average population belongs to normal weight category. The average age in our sample is 30 years. About 86% individuals are females and 73% individuals are married.

Table 1 presents the descriptive statistics grouped by overweight and obesity status. The mean difference across two groups with its statistical significance is also reported. Both average

blood glucose levels (both actual and ordinal values) and average diabetes prevalence (self-reported) are higher among overweight or obese individuals as compared to the non-overweight individuals. Average diabetes prevalence (self-reported) is three times among the overweight or obese individuals as compared to the non-overweight individuals. Mean Blood glucose levels are 10 mg/dl higher among overweight or obese individuals. The mean BMI among non-overweight individuals is 20.25 kg/m$^2$ which is lower than the mean for the total sample (21.71 kg/m$^2$), and the mean BMI among overweight or obese individuals is 28.32 kg/m$^2$. The average age of sample which is overweight or obese is 6 years higher than the non-overweight sample implying that the BMI tends to increase with age. Overweight or obese individuals' sample has higher averages for education, fried food and aerated drinks consumption, wealth quintile, and are more likely to be married and belong to urban regions as compared to the non-overweight individuals' sample. Also, overweight or obese individuals are less likely to belong to below poverty line households, scheduled caste and scheduled tribe.

Fig 1 illustrates BMI distribution amongst the diabetic and the non-diabetic population (as per self-reported diabetes status) for the entire sample. The solid red line represents BMI distribution for the diabetic population while dash-dotted blue line represents it for the non-diabetic population. It can be clearly seen that the BMI distribution for diabetic population lies to the right of the distribution for the non-diabetic population indicating that the diabetic population is more likely to have higher BMI. While around 46% of the diabetic population has BMI greater than 25, this proportion is only 18% amongst non-diabetic. We find similar results when we plotted BMI distributions by different categories of blood glucose levels (S1 Fig).

## Effect of Body Mass Index on the self-reported diabetes status: Probit and IV-Probit model estimates

Table 2 presents the average marginal effects of BMI on the self-reported diabetes status for the sample data that is restricted to married couples. Based on estimated Probit model, we compute the marginal effect of BMI on the self-reported diabetes status across overweight or obese, and non-overweight individuals. These marginal effects are reported for two classifications, for WHO International BMI classification in column (1) and for WHO Asian BMI classification in column (2). Within each column the average marginal effects of BMI, i.e., the change in probability of being diabetic due to a unit rise in BMI $\left(\frac{\partial P(D=1|X)}{\partial BMI}\right)$ is reported for overweight or obese individuals and non-overweight individuals along with the difference between the marginal effects across these two categories. Similarly, columns (3)–(4) report the results obtained from the IV-Probit model.

In all the model specifications, we include the same set of controls so that the marginal effects can be compared across different BMI categories. We control for demographic and socio-economic variables for individual and household characteristics, behavioural risk factors, eating habits and state fixed effects. Wald chi2 test statistic for both Probit and IV-Probit models along with their P-values are reported. For IV-Probit model, we use Wald test of exogeneity to check endogeneity of BMI. The rejection of the null hypothesis indicates that BMI is endogenous. We also report R$^2$ and F statistic for the first stage regression of the IV-Probit model as an approximate guide for the quality of our instrument. All the estimates are found to be robust to the inclusion or exclusion of controls.

Comparing Probit and IV-Probit model in each column, we find that marginal effects of BMI on self-reported diabetes status for IV-Probit model are substantially higher than those for the corresponding Probit model indicating that correlation estimates highly underestimate the casual effect of BMI on diabetes.

**Table 1. Descriptive statistics by overweight or obesity status.**

| Variable | Overweight or Obese | | Non-Overweight | | Difference[#] |
|---|---|---|---|---|---|
| | Mean | Standard Deviation | Mean | Standard Deviation | (t-statistic) |
| **Individual Characteristics** | | | | | |
| Self-Reported Diabetes Status | 0.037 | 0.189 | 0.010 | 0.097 | 0.027*** (78.578) |
| Ordinal Blood Glucose Levels | 0.154 | 0.443 | 0.051 | 0.245 | 0.103*** (1.2e+02) |
| Blood Glucose Levels–Actual Values (in mg/dl) | 113.571 | 42.203 | 102.723 | 25.586 | 10.848*** (1.3e+02) |
| Body Mass Index (in kg/m$^2$) | 28.317 | 3.323 | 20.249 | 2.493 | 8.068*** (1.1e+03) |
| Age (in years) | 35.282 | 8.705 | 28.910 | 9.859 | 6.372*** (2.3e+02) |
| Gender | 0.867 | 0.340 | 0.862 | 0.345 | 0.005*** (4.944) |
| Education | 1.625 | 0.983 | 1.451 | 0.994 | 0.174*** (60.889) |
| Marital Status | 0.897 | 0.304 | 0.695 | 0.460 | 0.202*** (1.6e+02) |
| Bank Account | 0.944 | 0.230 | 0.907 | 0.290 | 0.037*** (45.656) |
| Time since last ate (in hours) | 3.104 | 3.620 | 3.138 | 3.526 | -0.034*** (-3.335) |
| Time since last drink (in hours) | 4.031 | 10.141 | 5.685 | 14.761 | -1.654*** (-40.679) |
| **Behavioural Risk Factors** | | | | | |
| Smokes Cigarette | 0.025 | 0.156 | 0.024 | 0.153 | 0.001** (2.536) |
| Smokes Pipe | 0.0005 | 0.022 | 0.001 | 0.025 | -0.0002** (-2.392) |
| Chews Tobacco | 0.010 | 0.099 | 0.012 | 0.110 | -0.002*** (-7.352) |
| Snuffs | 0.001 | 0.033 | 0.001 | 0.034 | -0.000 (-0.345) |
| Smokes Cigar | 0.001 | 0.036 | 0.001 | 0.037 | -0.000 (-0.427) |
| Chews Paan or Gutkha | 0.039 | 0.192 | 0.051 | 0.221 | -0.013*** (-20.544) |
| Chews Paan with Tobacco | 0.045 | 0.207 | 0.043 | 0.203 | 0.002*** (3.325) |
| Drinks Alcohol | 0.062 | 0.241 | 0.065 | 0.247 | -0.003*** (-4.370) |
| **Eating Habits** | | | | | |
| Fried Food | 0.472 | 0.499 | 0.451 | 0.498 | 0.021*** (14.535) |
| Aerated Drinks | 0.280 | 0.449 | 0.234 | 0.423 | 0.046*** (37.081) |
| **Household Characteristics** | | | | | |
| Wealth Quintile | 2.745 | 1.207 | 1.814 | 1.363 | 0.930*** (2.4e+02) |
| Religion | 0.590 | 1.271 | 0.505 | 1.258 | 0.086*** (23.631) |
| Scheduled Caste | 0.151 | 0.358 | 0.187 | 0.390 | -0.036*** (-32.613) |
| Scheduled Tribe | 0.120 | 0.324 | 0.196 | 0.397 | -0.076*** (-68.823) |
| Other Backward Classes | 0.390 | 0.488 | 0.387 | 0.487 | 0.003** (2.353) |
| Insurance | 0.278 | 0.448 | 0.258 | 0.438 | 0.020*** (15.830) |
| Below Poverty Line | 0.286 | 0.452 | 0.408 | 0.491 | -0.122*** (-87.177) |
| Family Structure | 0.499 | 0.500 | 0.504 | 0.500 | -0.006*** (-4.089) |
| Number of Household Members | 5.531 | 2.696 | 5.825 | 2.638 | -0.294*** (-38.524) |
| Region | 0.452 | 0.498 | 0.257 | 0.437 | 0.195*** (1.5e+02) |

*** and ** indicates significance at 1% and 5% significance level.

[#] Difference = mean(Overweight or Obese)—mean(Non-Overweight). A positive value indicates that the mean is higher for overweight or obese population while a negative value indicates that the mean is higher for non-overweight population. The t-statistic is obtained from two-sample mean-comparison test with equal variances.

Comparing the marginal effects across overweight or obese individuals and non-overweight individuals in columns (1) and (2), based on Probit model, we find that the increase in the probability of being diabetic due to a unit rise in BMI is twice among overweight or obese individuals as compared to the non-overweight individuals. Whereas comparing the marginal effects across overweight or obese individuals and non-overweight individuals in columns (3) and (4), based on IV-Probit model, we find that the increase in the probability of being

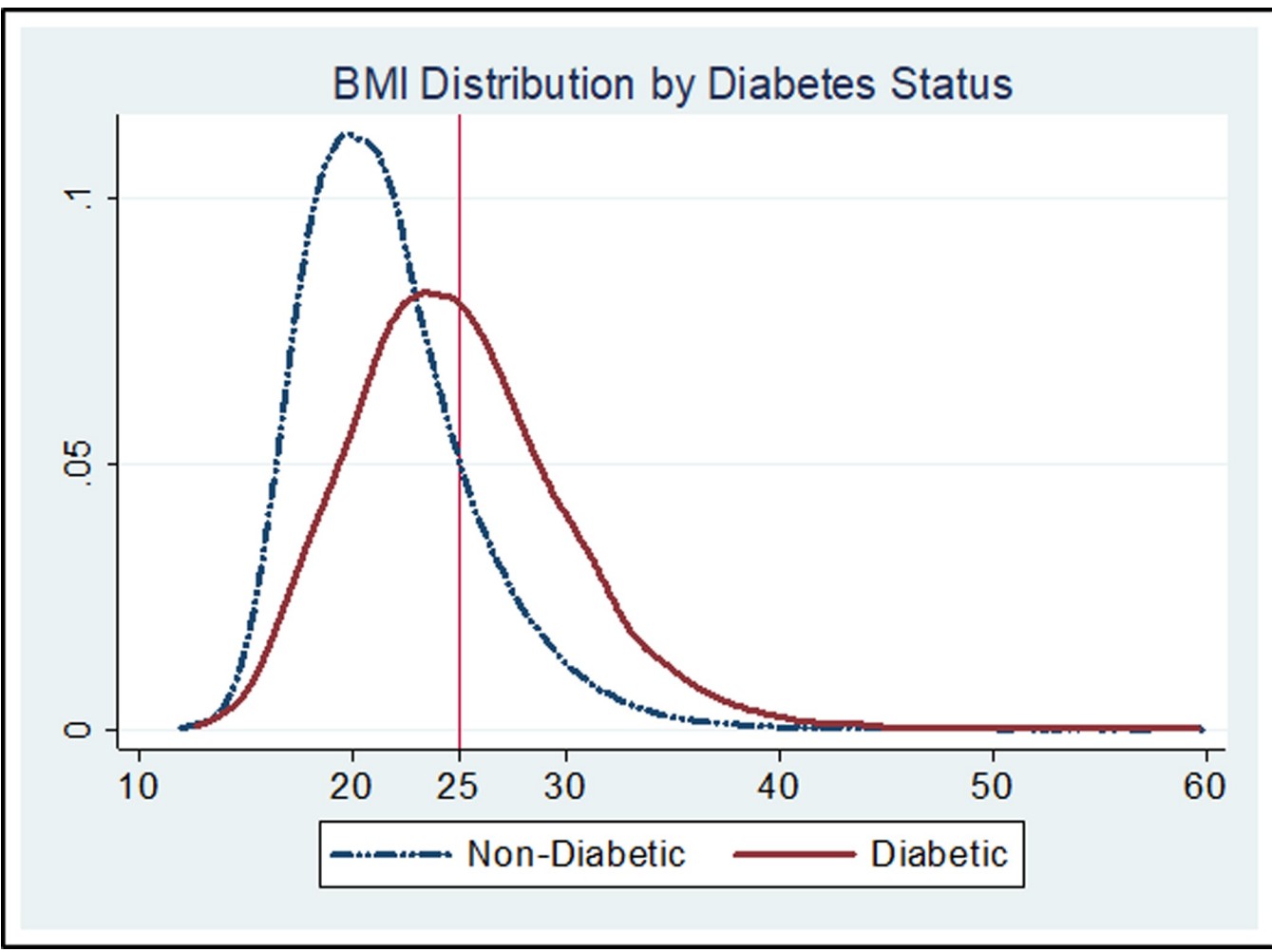

**Fig 1. BMI distribution by self-reported diabetes status.** Source: Figure constructed by author based on NFHS data for year 2015–16.

diabetic due to a unit rise in BMI is three times among overweight or obese individuals as compared to the non-overweight individuals. The marginal effect of BMI on the self-reported diabetes status for non-overweight individuals is 0.46% and for the overweight or obese individuals it is 1.48% for the IV-Probit model while the same figures for Probit model are 0.16% and 0.3%, respectively. We find that the marginal effects of BMI on the self-reported diabetes status differ significantly across non-overweight and overweight or obese individuals.

We also estimated the Probit model for the full sample data. These results are reported in S3 Table. The presentation of results in done in a similar fashion as explained for Table 2. Comparing the marginal effects across overweight or obese individuals and non-overweight individuals in columns (1) and (2), we find that the increase in the probability of being diabetic due to a unit rise in BMI is almost three times among overweight or obese individuals as compared to the non-overweight individuals. In column (1), the marginal effect of BMI on the self-reported diabetes status for non-overweight individuals is 0.08% and for the overweight or obese individuals it is 0.23%. Similar results are obtained by applying WHO Asian BMI classification, in column (2).

We next examine if the marginal effects of an increase in BMI on the likelihood of being diabetic differ across genders–male and female, regions–urban and rural, and wealth

**Table 2. Average marginal effects of BMI on self-reported diabetes status: Probit and IV-Probit model estimates for married couples sub-sample.**

| Marginal Effects | Probit Model | | IV-Probit Model | |
|---|---|---|---|---|
| | WHO International BMI Classification | WHO Asian BMI Classification | WHO International BMI Classification | WHO Asian BMI Classification |
| | (1) | (2) | (3) | (4) |
| Overweight or Obese Individuals | 0.0032*** (0.0004) | 0.0028*** (0.0003) | 0.0148*** (0.0038) | 0.0115*** (0.0028) |
| Non-Overweight Individuals | 0.0016*** (0.0001) | 0.0014*** (0.0001) | 0.0046*** (0.0008) | 0.0036*** (0.0005) |
| Difference[#] | 0.0016*** (0.0002) | 0.0014*** (0.0002) | 0.0101*** (0.0030) | 0.0079*** (0.0023) |
| Controls | Yes | | Yes | |
| State Fixed Effects | Yes | | Yes | |
| Observations | 43202 | | 43202 | |
| Wald chi2 | 1010.93 | | 234600.29 | |
| P-Value | 0.0000 | | 0.0000 | |
| Pseudo $R^2$ | 0.1072 | | | |
| Wald test of exogeneity, chi2 | | | 18.73 | |
| P-Value | | | 0.0000 | |
| First Stage | | | | |
| F–statistic | | | 153.28 | |
| $R^2$ | | | 0.2038 | |

*** represents significance at 1% significance level.

Delta-Method standard errors are reported in parentheses. "The delta method is used to estimate the standard errors of a non-linear function of model parameters (such as Ordered Probit, Probit or IV-Probit models). The delta method finds a linear approximation of the non-linear function to calculate the variance" [41].

[#] Difference is ME(Overweight and Obese)–ME(Non-Overweight).

Probit and IV-Probit models do not include marital status as a control. Marital status is omitted in the restricted sample as the sample comprises of only married individuals.

Controls include individual and household characteristics, behavioural risk factors and eating habits.

Individual and household characteristics include age, gender, education, bank account, household characteristics such as wealth quintile, religion, caste, insurance, below poverty line, family structure, number of household members and region.

Behavioural risk factors include smoking cigarette, smoking pipe, chewing tobacco, snuffing, smoking cigar, chewing paan or gutkha, chewing paan with tobacco and drinking alcohol.

Eating habits include daily or weekly consumption of fried foods and aerated drinks.

quintiles–poorest and richest. We do the analysis for the full sample comprising of married couples as well as for a sub-sample comprising of overweight or obese individuals within this set (having BMI $\geq 25$ kg/m$^2$). Table 3 presents the results obtained from Probit and IV-Probit model for the overweight and obese individuals sub-sample (within the restricted sample). The marginal effect of BMI on self-reported diabetes status is higher among men as compared to women in both specifications. However, the difference is not statistically significant. In both specifications, the increase in the likelihood of being diabetic with a unit increase in BMI in the urban population is about 1.3 times of the increase in the rural population. Also, the marginal effects in the richest wealth quintile are three times of those in the poorest wealth quintile in both the models (2.6 times $\approx$ 3 times in IV model). The marginal effects across regions and wealth quintiles differ statistically significantly. Similar results are obtained for the full sample comprising of married couples including both overweight or obese, and non-overweight individuals (S4 Table). The marginal effects are stronger for the sub-sample of overweight and obese individuals.

**Table 3. Average marginal effects of BMI on self-reported diabetes status amongst overweight or obese individuals (BMI $\geq$ 25 kg/m$^2$): Probit and IV-Probit model estimates for married couples sub-sample.**

| | Probit Model | | | IV-Probit Model | | |
|---|---|---|---|---|---|---|
| | Gender | Region | Wealth Quintile | Gender | Region | Wealth Quintile |
| | (1) | (2) | (3) | (4) | (5) | (6) |
| | Male | Urban | Richest | Male | Urban | Richest |
| Marginal Effects | 0.0026*** (0.0006) | 0.0031*** (0.0007) | 0.0035*** (0.0008) | 0.0175* (0.0093) | 0.0202** (0.0103) | 0.0221** (0.0108) |
| | Female | Rural | Poorest | Female | Rural | Poorest |
| Marginal Effects | 0.0019** (0.0009) | 0.0022*** (0.0005) | 0.0011*** (0.0003) | 0.0133 (0.0098) | 0.0152* (0.0085) | 0.0086 (0.0058) |
| | Difference# | Difference# | Difference# | Difference# | Difference# | Difference# |
| | 0.0007 (0.0009) | 0.0009*** (0.0003) | 0.0024*** (0.0006) | 0.0041 (0.0053) | 0.0049** (0.0020) | 0.0135** (0.0055) |
| Controls | Yes | | | Yes | | |
| State Fixed Effects | Yes | | | Yes | | |
| Observations | 9622 | | | 9711 | | |
| Wald chi2 | 394.56 | | | 106298.91 | | |
| P-Value | 0.0000 | | | 0.0000 | | |
| Pseudo R$^2$ | 0.1039 | | | | | |
| Wald test of exogeneity, chi2 | | | | 3.69 | | |
| P-Value | | | | 0.0547 | | |

***, ** and * represents significance at 1%, 5% and 10% significance level.

Delta-Method standard errors are reported in parentheses.

# (1) Difference is ME(Male)–ME(Female); (2) Difference is ME(Urban)–ME(Rural) and (3) Difference is ME(Richest)–ME(Poorest).

Probit and IV-Probit models do not include marital status as a control. Marital status is omitted in the restricted sample as the sample comprises of only married individuals.

Controls include individual and household characteristics, behavioural risk factors and eating habits.

Individual and household characteristics include age, gender, education, bank account, household characteristics such as wealth quintile, religion, caste, insurance, below poverty line, family structure, number of household members and region.

Behavioural risk factors include smoking cigarette, smoking pipe, chewing tobacco, snuffing, smoking cigar, chewing paan or gutkha, chewing paan with tobacco and drinking alcohol.

Eating habits include daily or weekly consumption of fried foods and aerated drinks.

### Effect of Body Mass Index on the blood glucose levels: Ordered Probit model estimates

Table 4 presents the average marginal effects of BMI on the ordinal blood glucose levels based on the Ordered Probit model estimation, i.e., the change in probability of belonging to a specific blood glucose category due to a unit rise in BMI $\left(\frac{\partial P(BG=j|X)}{\partial BMI}; j = 0, 1, 2\right)$ for the three blood glucose categories. We compute these effects across overweight or obese, and non-overweight individuals. While columns (1)–(3) report the marginal effects based on WHO International BMI classification, columns (4)–(6) report the results using WHO Asian BMI classification. Each column reports these marginal effects for the overweight or obese individuals, for the non-overweight individuals, and the difference across the two categories. Controls include the demographic and the socio-economic variables for individual and household characteristics, behavioural risk factors, eating habits and state fixed effects. We also control for the time since the individual last ate and drank (in hours) since these variables are expected to influence individual's blood glucose levels [42]. In the estimated model, the threshold coefficients, $\mu_0$ and $\mu_1$, are found to be positive, and $\mu_0 < \mu_1$. All the estimates are found to be robust to the inclusion or exclusion of controls.

**Table 4. Average marginal effects of BMI on ordinal blood glucose levels: Ordered Probit model estimates based on full sample data.**

| | Ordered Probit Model | | | | | |
| --- | --- | --- | --- | --- | --- | --- |
| | WHO International BMI Classification | | | WHO Asian BMI Classification | | |
| **Marginal Effects** | **Blood Glucose $\leq$ 140** | **141 $\leq$ Blood Glucose $\leq$ 200** | **Blood Glucose > 200** | **Blood Glucose $\leq$ 140** | **141 $\leq$ Blood Glucose $\leq$ 200** | **Blood Glucose > 200** |
| | **Normal Blood Glucose** | **Prediabetes** | **Diabetes** | **Normal Blood Glucose** | **Prediabetes** | **Diabetes** |
| | **(1)** | **(2)** | **(3)** | **(4)** | **(5)** | **(6)** |
| **Overweight or Obese** | -0.0068*** (0.0001) | 0.0048*** (0.00009) | 0.0020*** (0.00005) | -0.0061*** (0.0001) | 0.0044*** (0.00008) | 0.0017*** (0.00004) |
| **Non-Overweight Individuals** | -0.0035*** (0.00005) | 0.0027*** (0.00004) | 0.0007*** (0.00001) | -0.0031*** (0.00004) | 0.0025*** (0.00004) | 0.0006*** (0.00001) |
| **Difference#** | -0.0034*** (0.00008) | 0.0021*** (0.00005) | 0.0013*** (0.00003) | -0.0029*** (0.00007) | 0.0019*** (0.00004) | 0.0011*** (0.00003) |
| **Controls** | Yes | | | | | |
| **State Fixed Effects** | Yes | | | | | |
| **Observations** | 748,995 | | | | | |
| **Wald chi2** | 26968.90 | | | | | |
| **P-Value** | 0.0000 | | | | | |
| **Pseudo R$^2$** | 0.0901 | | | | | |

*** represents significance at 1% significance level.

Delta-Method standard errors are reported in parentheses.

# Difference is ME(Overweight and Obese) – ME(Non-Overweight).

Controls include individual and household characteristics, behavioural risk factors and eating habits.

Individual and household characteristics include age, gender, education, marital status, bank account, household characteristics such as wealth quintile, religion, caste, insurance, below poverty line, family structure, number of household members, region and time since last ate and drank.

Behavioural risk factors include smoking cigarette, smoking pipe, chewing tobacco, snuffing, smoking cigar, chewing paan or gutkha, chewing paan with tobacco and drinking alcohol.

Eating habits include daily or weekly consumption of fried foods and aerated drinks.

Comparing marginal effects across overweight or obese and non-overweight individuals reported in column (2), we find that the increase in the probability of being prediabetic due to a unit rise in BMI is almost twice among overweight or obese individuals as compared to the non-overweight individuals. The marginal effect of BMI on prediabetes for non-overweight individuals is 0.27% and for the overweight or obese individuals it is 0.48%. In column (3), the marginal effect of BMI on diabetes is 0.07% among non-overweight individuals which is 0.2% (about three times) among overweight or obese individuals. Similar results are obtained by applying WHO Asian BMI classification, in columns (5) and (6). The differences in the marginal effects are highly statistically significant.

Analogous to Tables 3 and 5 presents the results for the marginal effects of an increase in BMI on the ordinal blood glucose levels across genders, regions, and wealth quintiles for the overweight and obese individuals' sub-sample. With a unit increase in BMI, men are at a marginally higher risk of being both prediabetic (0.5%) and diabetic (0.3%) as compared to women (0.4% and 0.2% respectively). The marginal effects are moderately higher in the urban regions as compared to the rural regions. The individuals from the richest wealth quintile are 1.5 times more likely to be diabetic, and 1.2 times more likely to be prediabetic as compared to the poorest wealth quintile with a unit rise in BMI. The marginal effects across genders, regions and wealth quintiles differ statistically significantly. Similar results are obtained for the full sample (S5 Table). Again, the effects are stronger for the sub-sample of overweight and obese individuals.

**Table 5. Average marginal effects of BMI on ordinal blood glucose levels amongst overweight or obese individuals (BMI ≥ 25 kg/m²): Ordered Probit model estimates based on full sample data.**

| Marginal Effects | Ordered Probit Model | | |
|---|---|---|---|
| | Gender | | |
| | Male | Female | Difference# |
| Normal Blood Glucose (Blood Glucose ≤ 140) | -0.0078*** (0.0003) | -0.0064*** (0.0002) | -0.0014*** (0.0001) |
| Prediabetes (141 ≤ Blood Glucose ≤ 200) | 0.0046*** (0.0002) | 0.0041*** (0.0001) | 0.0005*** (0.00004) |
| Diabetes (Blood Glucose > 200) | 0.0032*** (0.0001) | 0.0023*** (0.00008) | 0.0009*** (0.00007) |
| | Region | | |
| | Urban | Rural | Difference# |
| Normal Blood Glucose (Blood Glucose ≤ 140) | -0.0070*** (0.0002) | -0.0062*** (0.0002) | -0.0008*** (0.00007) |
| Prediabetes (141 ≤ Blood Glucose ≤ 200) | 0.0043*** (0.0001) | 0.0040*** (0.0001) | 0.0003*** (0.00003) |
| Diabetes (Blood Glucose > 200) | 0.0026*** (0.0001) | 0.0022*** (0.00008) | 0.0005*** (0.00004) |
| | Wealth Quintile | | |
| | Richest | Poorest | Difference# |
| Normal Blood Glucose (Blood Glucose ≤ 140) | -0.0070*** (0.0002) | -0.0054*** (0.0002) | -0.0016*** (0.0002) |
| Prediabetes (141 ≤ Blood Glucose ≤ 200) | 0.0043*** (0.0001) | 0.0036*** (0.0001) | 0.0007*** (0.00008) |
| Diabetes (Blood Glucose > 200) | 0.0026*** (0.0001) | 0.0017*** (0.00009) | 0.0009*** (0.00008) |
| Controls | Yes | | |
| State Fixed Effects | Yes | | |
| Observations | 135,630 | | |
| Wald chi2 | 7482.14 | | |
| P-Value | 0.0000 | | |
| Pseudo R² | 0.0704 | | |

*** represents significance at 1% significance level.

Delta-Method standard errors are reported in parentheses.

# (1) Difference is ME(Male)–ME(Female); (2) Difference is ME(Urban)–ME(Rural) and (3) Difference is ME(Richest)–ME(Poorest).

Controls include individual and household characteristics, behavioural risk factors and eating habits.

Individual and household characteristics include age, gender, education, marital status, bank account, household characteristics such as wealth quintile, religion, caste, insurance, below poverty line, family structure, number of household members, region and time since last ate and drank.

Behavioural risk factors include smoking cigarette, smoking pipe, chewing tobacco, snuffing, smoking cigar, chewing paan or gutkha, chewing paan with tobacco and drinking alcohol.

Eating habits include daily or weekly consumption of fried foods and aerated drinks.

## Discussion

The study finds that an increase in BMI increases the risk of diabetes as well as prediabetes. Further this risk is higher amongst the overweight or obese individuals as compared to the non-overweight individuals. These results are in line with the studies by Sepp et al. [16] and Huffman et al. [4] which show that a rise in BMI is positively associated with the blood glucose levels and diabetes. The results obtained from our study are consistent across both WHO International, and Asian BMI classifications for defining overweight and obesity status of the population.

The change in probability of being prediabetic or diabetic with an additional unit gain in BMI is positive even for non-overweight individuals suggesting that weight gain increases the risk of diabetes regardless of individual being overweight or not. However, the level of risk varies with weight of an individual. The results are qualitatively similar for the two measures of diabetes, self-reported diabetes status and ordinal measure.

To examine how do the marginal effects vary across different subgroups with age and BMI, we plot the average marginal effects of BMI on diabetes. Fig 2 illustrates the graphical plot of the average marginal effects of BMI on self-reported diabetes status based on the Probit model estimates for the full sample data. We plot the average marginal effect of BMI on self-reported diabetes status for different values of age and BMI, and compare it across different subgroups–overweight or obese and non-overweight; male and female; and rural and urban. We find that the average marginal effect of BMI on self-reported diabetes status is considerably higher among overweight or obese individuals as compared to the non-overweight individuals. We do not witness any considerable difference in these marginal effects across gender. Also, the average marginal effects are higher for the urban population as compared to the rural population. It may be noted that the average marginal effect of BMI on self-reported diabetes status increases with both age and BMI across all subgroups.

Unlike other NCDs which mainly affect older age group population, diabetes affects younger age group population as well [5,4]. In India, 50.7% of the people who died from diabetes in 2017, died before the age of 60 years [43]. Diabetes also elevates the risk of other NCDs such as cardiovascular diseases, strokes, etc., and reduces health adjusted life expectancy [15,19]. The treatment of diabetes is expensive and is expected to impose an economic burden in the form of large increases in health care costs [44,45,46]. Since diabetes adversely affects health in many ways and has huge monetary burden associated with it, it is important to identify potential factors that have contributed to the rise in diabetes in India. Our study has identified overweight and obesity to be an important factor that has contributed to the increase in diabetes prevalence in India. The results suggest that to arrest the rising diabetes prevalence in India policy interventions that focus on reducing obesity levels would be helpful. Policies such as awareness campaigns, nutrition labelling on food products, restrictions on sale of aerated or high-sugar drinks, etc., may be effective in addressing the problem of obesity and diabetes prevalence in India.

A limitation of the study is we had to limit the sample data to married couples living in the same household of whom either is the head of the family for the IV-Probit specification as we use BMI of the spouse as an instrument. This restricts the generalisation of the results obtained from IV-Probit specification to married couples only. However, our correlation estimates can be generalised for the population at large in India. The other limitation is that we have cross section data. A panel or longitudinal data set could allow us to control for time-invariant individual fixed effects, and facilitate better understanding of how the effects of rising BMI on the likelihood of diabetes change overtime.

## Conclusion

Recognising the recently growing problem of overnutrition and diabetes in India, the study quantifies the causal effect of overweight and obesity on diabetes in India. The novel contribution of the study is that it addresses potential endogeneity problem while estimating the effect of BMI on diabetes. To the best of our knowledge this is the first study that addresses the role played by unobserved genetic and other related factors in the relationship between BMI and diabetes using an instrumental variable approach in the Indian context.

Considering two different health outcome variables–self-reported diabetes status and ordinal blood glucose levels, we examine the change in the likelihood of being diabetic and prediabetic with a rise in BMI across different subgroups of the population. We estimate IV-Probit, Probit and Ordered Probit models. We find that the marginal effect of BMI on diabetes is positive and statistically significant. Also, these effects are found to be much higher for the overweight or obese individuals as compared to the non-overweight individuals. Correlation

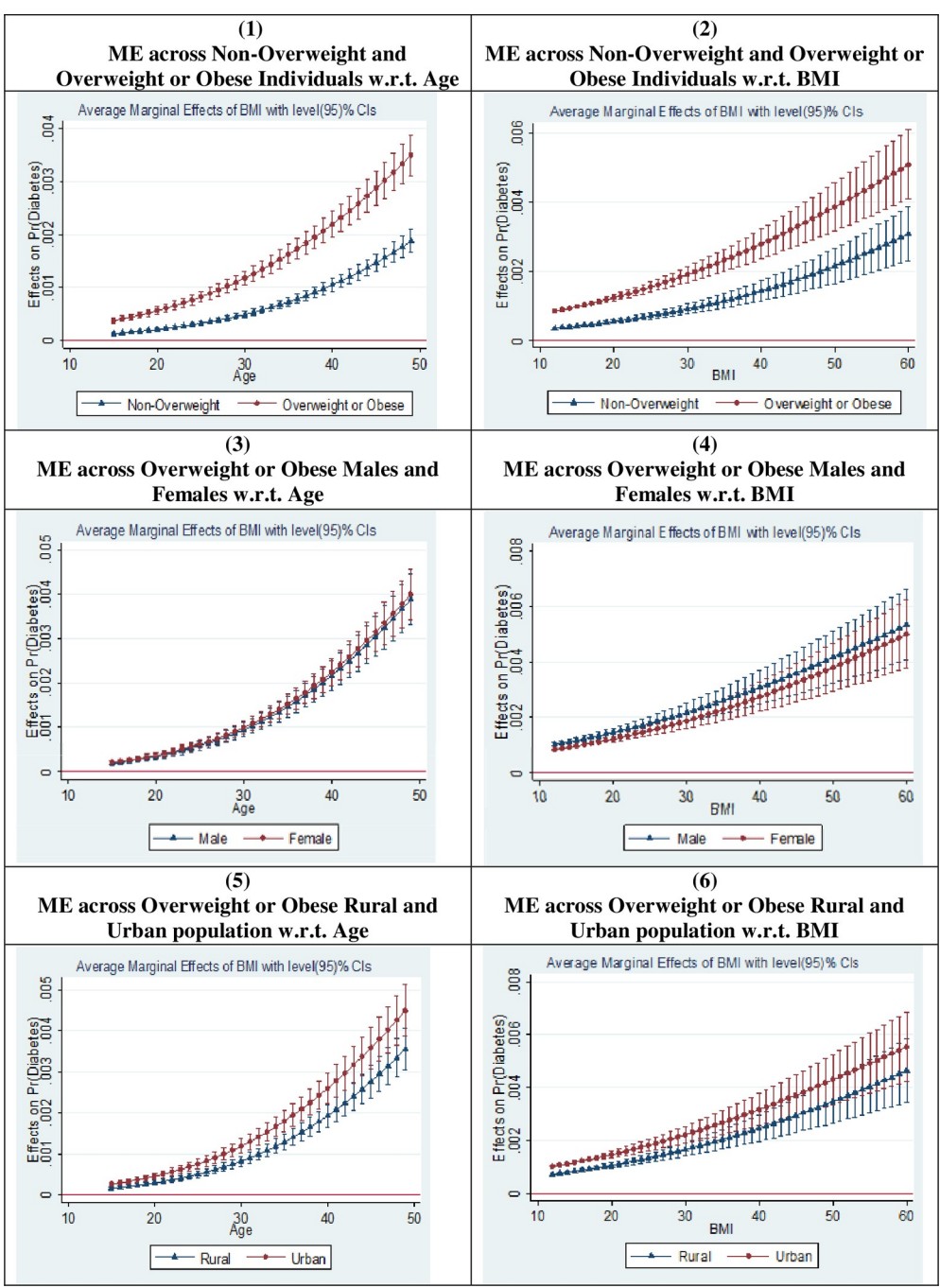

**Fig 2. Margins plot for the effect of BMI on the self-reported diabetes status.** Source: Figure constructed by authors. ME = Average marginal effect of BMI on self-reported diabetes status. In all graphs (1–6), the dark dot or triangle represents the average marginal effect of a unit rise in BMI on probability of being diabetic (measured on Y-axis). On X-axis, we have plotted either age or BMI (as labelled in each graph).

estimates highly understate the causal impact of the rise in BMI on diabetes. Heterogeneity analysis across different subgroups of the population suggests that among the overweight and obese individuals, men, population living in the urban areas and population belonging to the richest wealth quintile face a higher risk of being diabetic and prediabetic as compared to

women, population living in the rural areas, and population belonging to the poorest wealth quintile, respectively. Nonetheless, populations living in rural areas and belonging to lower wealth quintiles also face the risk of both diabetes and prediabetes.

Our findings have significant implications for the policy formulation as diabetes has a substantial health and economic burden associated with it. The cost burden associated with diabetes may have severe adverse impact on the households in India as more than 60% of the total health expenditure is financed by households privately in the form of out of pocket health expenditures (National Health Accounts (NHA)) [47]. It is concerning to note that contrary to the popular belief, risk of diabetes with an increase in BMI is not only restricted to urban areas but also in rural areas and is no longer a disease of the rich. Our results indicate that population belonging to the poorest wealth quintiles also face a substantial risk of diabetes. Diabetes among poor households may have catastrophic implications and lead to extreme impoverishment. Therefore, there is an urgent need to have policies that address rising overweight and obesity prevalence in India.

Future research may examine the effects of overnutrition on other NCDs such as cardiovascular diseases, hypertension, etc. Researchers may also quantify the health care burden associated with diabetes.

## Supporting information

**S1 Appendix. Statistical analytical codes used in the analysis.**
(DOCX)

**S1 Fig. BMI distribution by blood glucose levels.** Source: Figure constructed by author based on NFHS data for year 2015–16. Blood glucose levels are measured in mg/dl.
(DOCX)

**S1 Table. List of variables with definition and type.**
(DOCX)

**S2 Table. Descriptive statistics.**
(DOCX)

**S3 Table. Average marginal effects of BMI on self-reported diabetes status: Probit model estimates based on full sample data.**
(DOCX)

**S4 Table. Average marginal effects of BMI on self-reported diabetes status: Probit and IV-Probit model estimates for married couples sub-sample.**
(DOCX)

**S5 Table. Average marginal effects of BMI on ordinal blood glucose levels: Ordered Probit model estimates based on full sample data.**
(DOCX)

## Acknowledgments

The authors thank all the participants present at the Economics Seminar, Centre for International Trade and Development, School of International Studies, Jawaharlal Nehru University, New Delhi, India and at 2nd International Conference on Business, Economics & Sustainable Development at TERI School of Advanced Studies, New Delhi, India where this paper was presented. This research is a part of PhD Thesis of Shivani Gupta under the supervision of Professor Sangeeta Bansal.

## Author Contributions

**Conceptualization:** Shivani Gupta, Sangeeta Bansal.

**Formal analysis:** Shivani Gupta, Sangeeta Bansal.

**Methodology:** Shivani Gupta, Sangeeta Bansal.

**Supervision:** Sangeeta Bansal.

**Writing – original draft:** Shivani Gupta.

**Writing – review & editing:** Shivani Gupta, Sangeeta Bansal.

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
