## [Decision Letter · Decision Letter 0]

6 Dec 2019

PONE-D-19-24183

Does a rise in BMI causes an increased risk of diabetes?: Evidence from India

PLOS ONE

Dear Ms. Gupta,

Thank you for submitting your manuscript to PLOS ONE. After careful consideration, we feel that it has merit but does not fully meet PLOS ONE’s publication criteria as it currently stands. Therefore, we invite you to submit a revised version of the manuscript that addresses the points raised during the review process.

We would appreciate receiving your revised manuscript by Jan 20 2020 11:59PM. To enhance the reproducibility of your results, we recommend that if applicable you deposit your laboratory protocols in protocols.io, where a protocol can be assigned its own identifier (DOI) such that it can be cited independently in the future. For instructions see: http://journals.plos.org/plosone/s/submission-guidelines#loc-laboratory-protocols

We look forward to receiving your revised manuscript.

Kind regards,

Petri Böckerman

Academic Editor

PLOS ONE

Journal Requirements:

2. In ethics statement in the manuscript and in the online submission form, please provide additional information about the patient records used in your retrospective study. Specifically, please ensure that you have discussed whether all data were fully anonymized before you accessed them and/or whether the IRB or ethics committee waived the requirement for informed consent. If patients provided informed written consent to have data from their medical records used in research, please include this information.

3. We noticed you refer several times to a "causal" relationship in your manuscript. Please consider carefully whether or not this study allows for such conclusions.

Reviewers' comments:

Reviewer's Responses to Questions

**Comments to the Author**

1. Is the manuscript technically sound, and do the data support the conclusions?

Reviewer #1: Yes

Reviewer #2: Yes

2. Has the statistical analysis been performed appropriately and rigorously? 

Reviewer #1: Yes

Reviewer #2: Yes

3. Have the authors made all data underlying the findings in their manuscript fully available?

Reviewer #1: Yes

Reviewer #2: Yes

4. Is the manuscript presented in an intelligible fashion and written in standard English?

Reviewer #1: Yes

Reviewer #2: Yes

5. Review Comments to the Author

Reviewer #1: I congratulate authors for the nicely written manuscript. The hypothesis's are well defined and presented. However, there are some minor suggestion for improvising the manuscript.

General:

Grammar needs slight improvisation; in terms of tenses used in the manuscript.

Introduction

Page 5; 3rd Paragraph; 6th Line: "We convert the reported blood glucose levels into an ordinal measure by dividing it into three categories": Authors are requested to either specify the categories here in the text or mention that the ordinal classification is elaborated later in Methods section.

Page 6: 2nd Paragraph; Individuals with diabetes are less likely to report having a good health as compared to the non-diabetic individuals; Authors may provide reference.

Page 6: 2nd Paragraph: There is repetition in the manuscript, which is adding to the length of the chapter. The authors have time and again highlighted that ‘The findings of this paper have policy implications...’ (Page 6, Page 35). It would be in the interest of the paper that the language is kept crisp and to the point.

Methods

Page 7: ‘We test the third and fourth hypotheses for a sub-sample comprising of overweight or obese population as they are expected to be facing a higher risk of diabetes.’ Not sure what do authors mean by this. Hypothesis 3 and 4 could have been tested for the entire group (not only the sub sample).

Reviewer #2: Present study examines whether rise in BMI is associated with an increased risk of diabetes in a large household data from India.

Main strengths of the present study are strong data and analytical methods (the use of IV-analysis).

The study is, however, difficult to follow as it contains so much information that is not present in a best possible way.

I have the following suggestions how the present study could be improved.

Introduction

Introduction could focus only on the study background factors and previous evidence. The study aims could be outlined in the last chapter of the introduction. Methodological details could be moved to methods and potential policy implications to discussion.

Plos One does not permit footnotes so they should be removed and the information included to the text, if needed.

Methods

From this section, hypothesis should be moved to the introduction. Separate the subchapter for data, measurements, and statistical analyses, preferably in this order, could be created. Descriptive statistics and their interpretation could be moved to the first chapter in results.

Results

The results contain multiple tables, could some of these moved to online supplement appendix?

Discussion

In the present format, discussion is rather limited. Potential policy implications of the present findings should be discussed. Similarly it would be important to discuss generalizability of the present findings. Potential limitations of the instrument variable should be discussed.

Could the statistical analytical code be included in the online appendix?

6. PLOS authors have the option to publish the peer review history of their article (what does this mean?). If published, this will include your full peer review and any attached files.

Reviewer #1: No

Reviewer #2: No

---

## [Author Response · Author response to Decision Letter 0]

4 Feb 2020

Response to the Reviewers of the Manuscript PONE-D-19-24183 “Does a rise in BMI causes an increased risk of diabetes?: Evidence from India”

We thank the reviewers for their time and effort in reviewing our manuscript and providing us with very useful suggestions. We have thoroughly revised the write-up and presentation of the text in the manuscript incorporating all the suggestions. This has vastly improved the revised manuscript. The revised manuscript does not have repetitions, is crisp, has a better focus, and is considerably smaller in size.

Reviewer #1:

Comment:

General: Grammar needs slight improvisation; in terms of tenses used in the manuscript.

Response:

Manuscript has been thoroughly revised and all repetitive statements have been removed. We have tried our best to keep the language crisp. Total pages in the manuscript have reduced from 40 to 31. Table 1, 2 and 6 (as per the first draft of the manuscript) have been moved to online supplement appendix and Table 4 (as per the first draft of the manuscript) has been removed to reduce the length of the manuscript.

Comment:

Introduction

Page 5; 3rd Paragraph; 6th Line: "We convert the reported blood glucose levels into an ordinal measure by dividing it into three categories": Authors are requested to either specify the categories here in the text or mention that the ordinal classification is elaborated later in Methods section.

Response:

We have moved this information to ‘Methods’ section along with the variable definition (Page 6; 2nd Paragraph).

Comment:

Page 6: 2nd Paragraph; Individuals with diabetes are less likely to report having a good health as compared to the non-diabetic individuals; Authors may provide reference.

Response:

To make the introduction section crisp, this statement has been removed. However, the reference is:

Graue, M., Wentzel-Larsen, T., Hanestad, B., Båtsvik, B., and Søvik, O. (2003). Measuring Self-Reported, Health-Related, Quality of Life in Adolescents with Type 1 Diabetes using Both Generic and Disease-Specific Instruments. Acta paediatrica, 92(10): 1190-6.

Comment:

Page 6: 2nd Paragraph: There is repetition in the manuscript, which is adding to the length of the chapter. The authors have time and again highlighted that ‘The findings of this paper have policy implications...’ (Page 6, Page 35). It would be in the interest of the paper that the language is kept crisp and to the point.

Response:

The repeated statements have been removed and policy implications are now discussed only in the discussion section (Page 23; 4th Paragraph). 

Comment:

Methods

Page 7: ‘We test the third and fourth hypotheses for a sub-sample comprising of overweight or obese population as they are expected to be facing a higher risk of diabetes.’ Not sure what do authors mean by this. Hypothesis 3 and 4 could have been tested for the entire group (not only the sub sample).

Response:

We have tested the hypothesis 3 and 4 for the entire group and have provided complete result tables in the supporting information (S4 and S5 Table). Similar results are obtained for the entire sample and sub-sample of overweight and obese individuals with latter experiencing stronger effects of a unit rise in BMI on the probability of being diabetic or prediabetic. 

Reviewer #2:

Comment:

Introduction

Introduction could focus only on the study background factors and previous evidence. The study aims could be outlined in the last chapter of the introduction. Methodological details could be moved to methods and potential policy implications to discussion.

Response:

The Introduction has been revised. It now presents the background factors and previous evidence for the study.

Last para of introduction now outlines the aim and hypothesis of the study (Page 5).

Methodological details have been moved to methods (Page 6) and potential policy implications to discussion (Page 23; 4th Paragraph).

Comment:

Plos One does not permit footnotes so they should be removed and the information included to the text, if needed.

Response:

All the footnotes have been removed and the relevant information has been moved to the text.

Comment:

Methods

From this section, hypothesis should be moved to the introduction. Separate the subchapter for data, measurements, and statistical analyses, preferably in this order, could be created. Descriptive statistics and their interpretation could be moved to the first chapter in results.

Response:

Hypotheses have been moved to the introduction (Page 5). Separate subchapter made for data (Page 11). Statistical analyses in presented in two subchapetrs, one, for IV-Probit model (Page 8) and, the other, for Ordered Probit model (Page 10). Descriptive statistics and their interpretation has been moved to the first chapter in results (Page 11).

Comment:

Results

The results contain multiple tables, could some of these moved to online supplement appendix?

Response:

Tables 1, 2 and 6, as per the first draft of the manuscript have been moved to online supplement appendix and are provided as supporting information (S1, S2 and S3 Table respectively, as per the revised draft of the manuscript). Table 4 (as per the first draft of the manuscript) has been removed to reduce length of the manuscript and avoid repetition. 

Comment:

Discussion

In the present format, discussion is rather limited. Potential policy implications of the present findings should be discussed. Similarly it would be important to discuss generalizability of the present findings. Potential limitations of the instrument variable should be discussed.

Response:

Dicussion section has been revised to include policy implications (Page 23; 4th Paragraph) and potential limitations of the study (Page 24; 2nd Paragraph). Generalizability of the present findings (Page 24; 2nd Paragraph) and potential limitations of the instrument variable are discussed.

Comment:

Could the statistical analytical code be included in the online appendix?

Response:

Statistical analytical code has been provided in the online appendix (S1 Appendix).

Further the corresponding author has changed from Shivani Gupta to Sangeeta Bansal.

---

## [Editor Report · Decision Letter 1]

13 Feb 2020

Does a rise in BMI causes an increased risk of diabetes?: Evidence from India

PONE-D-19-24183R1

Dear Dr. Bansal,

We are pleased to inform you that your manuscript has been judged scientifically suitable for publication and will be formally accepted for publication once it complies with all outstanding technical requirements.

With kind regards,

Petri Böckerman

Academic Editor

PLOS ONE
---

## [Editor Report · Acceptance letter]

3 Mar 2020

PONE-D-19-24183R1 

Does a rise in BMI causes an increased risk of diabetes?: Evidence from India 

Dear Dr. Bansal:

I am pleased to inform you that your manuscript has been deemed suitable for publication in PLOS ONE. Congratulations! Your manuscript is now with our production department. 

With kind regards,

on behalf of

Professor Petri Böckerman 

Academic Editor

PLOS ONE